# Learning Non-Convergent Non-Persistent Short-Run MCMC Toward Energy-Based Model

**Erik Nijkamp**
UCLA Department of Statistics
enijkamp@ucla.edu

**Mitch Hill**
UCLA Department of Statistics
mkhill@ucla.edu

**Song-Chun Zhu**
UCLA Department of Statistics
sczhu@stat.ucla.edu

**Ying Nian Wu**
UCLA Department of Statistics
ywu@stat.ucla.edu

## Abstract

This paper studies a curious phenomenon in learning energy-based model (EBM) using MCMC. In each learning iteration, we generate synthesized examples by running a non-convergent, non-mixing, and non-persistent short-run MCMC toward the current model, always starting from the same initial distribution such as uniform noise distribution, and always running a fixed number of MCMC steps. After generating synthesized examples, we then update the model parameters according to the maximum likelihood learning gradient, as if the synthesized examples are fair samples from the current model. We treat this non-convergent short-run MCMC as a learned generator model or a flow model. We provide arguments for treating the learned non-convergent short-run MCMC as a valid model. We show that the learned short-run MCMC is capable of generating realistic images. More interestingly, unlike traditional EBM or MCMC, the learned short-run MCMC is capable of reconstructing observed images and interpolating between images, like generator or flow models. The code can be found in the Appendix.

## 1 Introduction

### 1.1 Learning Energy-Based Model by MCMC Sampling

The maximum likelihood learning of the energy-based model (EBM) [32, 55, 22, 44, 33, 37, 8, 35, 52, 53, 25, 9, 51] follows what Grenander [17] called "analysis by synthesis" scheme. Within each learning iteration, we generate synthesized examples by sampling from the current model, and then update the model parameters based on the difference between the synthesized examples and the observed examples, so that eventually the synthesized examples match the observed examples in terms of some statistical properties defined by the model. To sample from the current EBM, we need to use Markov chain Monte Carlo (MCMC), such as the Gibbs sampler [14], Langevin dynamics, or Hamiltonian Monte Carlo [36]. Recent work that parametrizes the energy function by modern convolutional neural networks (ConvNets) [31, 29] suggests that the "analysis by synthesis" process can indeed generate highly realistic images [52, 13, 24, 12].

Although the "analysis by synthesis" learning scheme is intuitively appealing, the convergence of MCMC can be impractical, especially if the energy function is multi-modal, which is typically the case if the EBM is to approximate the complex data distribution, such as that of natural images. For such EBM, the MCMC usually does not mix, i.e., MCMC chains from different starting points tend to get trapped in different local modes instead of traversing modes and mixing with each other.

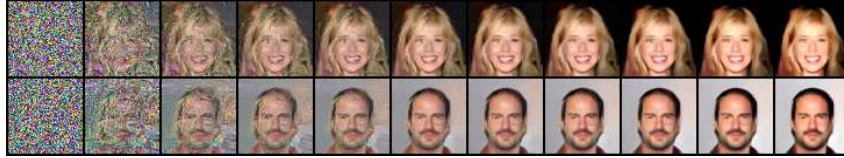

Figure 1: **Synthesis by short-run MCMC**: Generating synthesized examples by running 100 steps of Langevin dynamics initialized from uniform noise for CelebA ($64 \times 64$).

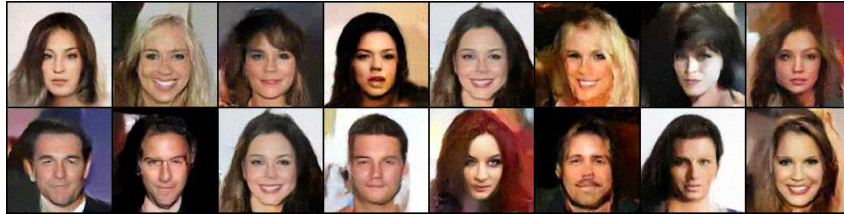

Figure 2: **Synthesis by short-run MCMC**: Generating synthesized examples by running 100 steps of Langevin dynamics initialized from uniform noise for CelebA ($128 \times 128$).

## 1.2 Short-Run MCMC as Generator or Flow Model

In this paper, we investigate a learning scheme that is apparently wrong with no hope of learning a valid model. Within each learning iteration, we run a non-convergent, non-mixing and non-persistent short-run MCMC, such as 5 to 100 steps of Langevin dynamics, toward the current EBM. Here, we always initialize the non-persistent short-run MCMC from the same distribution, such as the uniform noise distribution, and we always run the same number of MCMC steps. We then update the model parameters as usual, as if the synthesized examples generated by the non-convergent and non-persistent noise-initialized short-run MCMC are the fair samples generated from the current EBM. We show that, after the convergence of such a learning algorithm, the resulting noise-initialized short-run MCMC can generate realistic images, see Figures 1 and 2.

The short-run MCMC is not a valid sampler of the EBM because it is short-run. As a result, the learned EBM cannot be a valid model because it is learned based on a wrong sampler. Thus we learn a wrong sampler of a wrong model. However, the short-run MCMC can indeed generate realistic images. What is going on?

The goal of this paper is to understand the learned short-run MCMC. We provide arguments that it is a valid model for the data in terms of matching the statistical properties of the data distribution. We also show that the learned short-run MCMC can be used as a generative model, such as a generator model [15, 28] or the flow model [10, 11, 27, 5, 16], with the Langevin dynamics serving as a noise-injected residual network, with the initial image serving as the latent variables, and with the initial uniform noise distribution serving as the prior distribution of the latent variables. We show that unlike traditional EBM and MCMC, the learned short-run MCMC is capable of reconstructing the observed images and interpolating different images, just like a generator or a flow model can do. See Figures 3 and 4. This is very unconventional for EBM or MCMC, and this is due to the fact that the MCMC is non-convergent, non-mixing and non-persistent. In fact, our argument applies to the situation where the short-MCMC does not need to have the EBM as the stationary distribution.

While the learned short-run MCMC can be used for synthesis, the above learning scheme can be generalized to tasks such as image inpainting, super-resolution, style transfer, or inverse optimal control [56, 2] etc., using informative initial distributions and conditional energy functions.

## 2 Contributions and Related Work

This paper constitutes a conceptual shift, where we shift attention from learning EBM with unrealistic convergent MCMC to the non-convergent short-run MCMC. This is a break away from the long tradition of both EBM and MCMC. We provide theoretical and empirical evidence that the learned short-run MCMC is a valid generator or flow model. This conceptual shift frees us from the convergence issue of MCMC, and makes the short-run MCMC a reliable and efficient technology.

More generally, we shift the focus from energy-based model to energy-based dynamics. This appears to be consistent with the common practice of computational neuroscience [30], where researchers often directly start from the dynamics, such as attractor dynamics [23, 3, 40] whose express goal is to

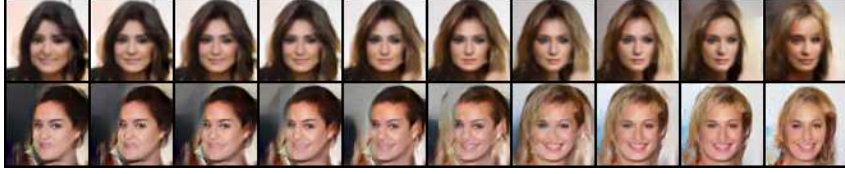

Figure 3: **Interpolation by short-run MCMC resembling a generator or flow model**: The transition depicts the sequence $M_\theta(z_\rho)$ with interpolated noise $z_\rho = \rho z_1 + \sqrt{1-\rho^2}z_2$ where $\rho \in [0,1]$ on CelebA ($64 \times 64$). Left: $M_\theta(z_1)$. Right: $M_\theta(z_2)$. See Section 3.4.

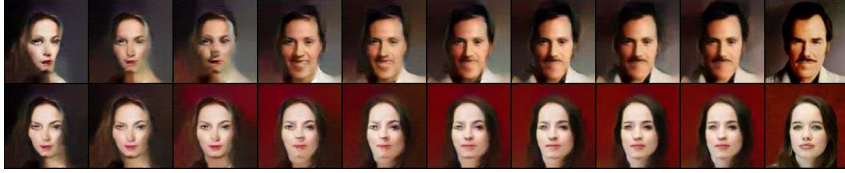

Figure 4: **Reconstruction by short-run MCMC resembling a generator or flow model**: The transition depicts $M_\theta(z_t)$ over time $t$ from random initialization $t=0$ to reconstruction $t=200$ on CelebA ($64 \times 64$). Left: Random initialization. Right: Observed examples. See Section 3.4.

be trapped in a local mode. It is our hope that our work may help to understand the learning of such dynamics. We leave it to future work.

For short-run MCMC, contrastive divergence (CD) [21] is the most prominent framework for theoretical underpinning. The difference between CD and our study is that in our study, the short-run MCMC is initialized from noise, while CD initializes from observed images. CD has been generalized to persistent CD [48]. Compared to persistent MCMC, the non-persistent MCMC in our method is much more efficient and convenient. [38] performs a thorough investigation of various persistent and non-persistent, as well as convergent and non-convergent learning schemes. In particular, the emphasis is on learning proper energy function with persistent and convergent Markov chains. In all of the CD-based frameworks, the goal is to learn the EBM, whereas in our framework, we discard the learned EBM, and only keep the learned short-run MCMC.

Our theoretical understanding of short-run MCMC is based on generalized moment matching estimator. It is related to moment matching GAN [34], however, we do not learn a generator adversarially.

## 3 Non-Convergent Short-Run MCMC as Generator Model

### 3.1 Maximum Likelihood Learning of EBM

Let $x$ be the signal, such as an image. The energy-based model (EBM) is a Gibbs distribution

$$p_\theta(x) = \frac{1}{Z(\theta)} \exp(f_\theta(x)), \tag{1}$$

where we assume $x$ is within a bounded range. $f_\theta(x)$ is the negative energy and is parametrized by a bottom-up convolutional neural network (ConvNet) with weights $\theta$. $Z(\theta) = \int \exp(f_\theta(x))dx$ is the normalizing constant.

Suppose we observe training examples $x_i, i=1,...,n \sim p_{\text{data}}$, where $p_{\text{data}}$ is the data distribution. For large $n$, the sample average over $\{x_i\}$ approximates the expectation with respect with $p_{\text{data}}$. For notational convenience, we treat the sample average and the expectation as the same.

The log-likelihood is

$$L(\theta) = \frac{1}{n}\sum_{i=1}^{n} \log p_\theta(x_i) \doteq \mathbb{E}_{p_{\text{data}}}[\log p_\theta(x)]. \tag{2}$$

The derivative of the log-likelihood is

$$L'(\theta) = \mathbb{E}_{p_{\text{data}}}\left[\frac{\partial}{\partial \theta}f_\theta(x)\right] - \mathbb{E}_{p_\theta}\left[\frac{\partial}{\partial \theta}f_\theta(x)\right] \doteq \frac{1}{n}\sum_{i=1}^{n}\frac{\partial}{\partial \theta}f_\theta(x_i) - \frac{1}{n}\sum_{i=1}^{n}\frac{\partial}{\partial \theta}f_\theta(x_i^-), \tag{3}$$

where $x_i^- \sim p_\theta(x)$ for $i = 1,...,n$ are the generated examples from the current model $p_\theta(x)$.

The above equation leads to the "analysis by synthesis" learning algorithm. At iteration $t$, let $\theta_t$ be the current model parameters. We generate $x_i^- \sim p_{\theta_t}(x)$ for $i = 1,...,n$. Then we update $\theta_{t+1} = \theta_t + \eta_t L'(\theta_t)$, where $\eta_t$ is the learning rate.

## 3.2 Short-Run MCMC

Generating synthesized examples $x_i^- \sim p_\theta(x)$ requires MCMC, such as Langevin dynamics (or Hamiltonian Monte Carlo) [36], which iterates

$$x_{\tau+\Delta\tau} = x_\tau + \frac{\Delta\tau}{2} f'_\theta(x_\tau) + \sqrt{\Delta\tau} U_\tau, \tag{4}$$

where $\tau$ indexes the time, $\Delta\tau$ is the discretization of time, and $U_\tau \sim N(0, I)$ is the Gaussian noise term. $f'_\theta(x) = \partial f_\theta(x)/\partial x$ can be obtained by back-propagation. If $p_\theta$ is of low entropy or low temperature, the gradient term dominates the diffusion noise term, and the Langevin dynamics behaves like gradient descent.

If $f_\theta(x)$ is multi-modal, then different chains tend to get trapped in different local modes, and they do not mix. We propose to give up the sampling of $p_\theta$. Instead, we run a fixed number, e.g., $K$, steps of MCMC, toward $p_\theta$, starting from a fixed initial distribution, $p_0$, such as the uniform noise distribution. Let $M_\theta$ be the $K$-step MCMC transition kernel. Define

$$q_\theta(x) = (M_\theta p_0)(z) = \int p_0(z) M_\theta(x|z) dz, \tag{5}$$

which is the marginal distribution of the sample $x$ after running $K$-step MCMC from $p_0$.

In this paper, instead of learning $p_\theta$, we treat $q_\theta$ to be the target of learning. After learning, we keep $q_\theta$, but we discard $p_\theta$. That is, the sole purpose of $p_\theta$ is to guide a $K$-step MCMC from $p_0$.

## 3.3 Learning Short-Run MCMC

The learning algorithm is as follows. Initialize $\theta_0$. At learning iteration $t$, let $\theta_t$ be the model parameters. We generate $x_i^- \sim q_{\theta_t}(x)$ for $i = 1,...,m$. Then we update $\theta_{t+1} = \theta_t + \eta_t \Delta(\theta_t)$, where

$$\Delta(\theta) = \mathbb{E}_{p_{\text{data}}} \left[ \frac{\partial}{\partial\theta} f_\theta(x) \right] - \mathbb{E}_{q_\theta} \left[ \frac{\partial}{\partial\theta} f_\theta(x) \right] \approx \sum_{i=1}^m \frac{\partial}{\partial\theta} f_\theta(x_i) - \sum_{i=1}^m \frac{\partial}{\partial\theta} f_\theta(x_i^-). \tag{6}$$

We assume that the algorithm converges so that $\Delta(\theta_t) \to 0$. At convergence, the resulting $\theta$ solves the estimating equation $\Delta(\theta) = 0$.

To further improve training, we smooth $p_{\text{data}}$ by convolution with a Gaussian white noise distribution, i.e., injecting additive noises $\varepsilon_i \sim N(0, \sigma^2 I)$ to observed examples $x_i \leftarrow x_i + \varepsilon_i$ [46, 43]. This makes it easy for $\Delta(\theta_t)$ to converge to 0, especially if the number of MCMC steps, $K$, is small, so that the estimating equation $\Delta(\theta) = 0$ may not have solution without smoothing $p_{\text{data}}$.

The learning procedure in Algorithm 1 is simple. The key to the above algorithm is that the generated $\{x_i^-\}$ are independent and fair samples from the model $q_\theta$.

---

**Algorithm 1:** Learning short-run MCMC. See code in Appendix 7.3.

---

**input** : Negative energy $f_\theta(x)$, training steps $T$, initial weights $\theta_0$, observed examples $\{x_i\}_{i=1}^n$, batch size $m$, variance of noise $\sigma^2$, Langevin descretization $\Delta\tau$ and steps $K$, learning rate $\eta$.

**output** : Weights $\theta_{T+1}$.

**for** $t = 0 : T$ **do**

    1. Draw observed images $\{x_i\}_{i=1}^m$.
    2. Draw initial negative examples $\{x_i^-\}_{i=1}^m \sim p_0$.
    3. Update observed examples $x_i \leftarrow x_i + \varepsilon_i$ where $\varepsilon_i \sim N(0, \sigma^2 I)$.
    4. Update negative examples $\{x_i^-\}_{i=1}^m$ for $K$ steps of Langevin dynamics (4).
    5. Update $\theta_t$ by $\theta_{t+1} = \theta_t + g(\Delta(\theta_t), \eta, t)$ where gradient $\Delta(\theta_t)$ is (6) and $g$ is ADAM [26].

---

### 3.4  Generator or Flow Model for Interpolation and Reconstruction

We may consider $q_\theta(x)$ to be a generative model,

$$z \sim p_0(z); \; x = M_\theta(z, u), \tag{7}$$

where $u$ denotes all the randomness in the short-run MCMC. For the $K$-step Langevin dynamics, $M_\theta$ can be considered a $K$-layer noise-injected residual network. $z$ can be considered latent variables, and $p_0$ the prior distribution of $z$. Due to the non-convergence and non-mixing, $x$ can be highly dependent on $z$, and $z$ can be inferred from $x$. This is different from the convergent MCMC, where $x$ is independent of $z$. When the learning algorithm converges, the learned EBM tends to have low entropy and the Langevin dynamics behaves like gradient descent, where the noise terms are disabled, i.e., $u = 0$. In that case, we simply write $x = M_\theta(z)$.

We can perform interpolation as follows. Generate $z_1$ and $z_2$ from $p_0(z)$. Let $z_\rho = \rho z_1 + \sqrt{1 - \rho^2} z_2$. This interpolation keeps the marginal variance of $z_\rho$ fixed. Let $x_\rho = M_\theta(z_\rho)$. Then $x_\rho$ is the interpolation of $x_1 = M_\theta(z_1)$ and $x_2 = M_\theta(z_2)$. Figure 3 displays $x_\rho$ for a sequence of $\rho \in [0, 1]$.

For an observed image $x$, we can reconstruct $x$ by running gradient descent on the least squares loss function $L(z) = \|x - M_\theta(z)\|^2$, initializing from $z_0 \sim p_0(z)$, and iterates $z_{t+1} = z_t - \eta_t L'(z_t)$. Figure 4 displays the sequence of $x_t = M_\theta(z_t)$.

In general, $z \sim p_0(z); x = M_\theta(z, u)$ defines an energy-based dynamics. $K$ does not need to be fixed. It can be a stopping time that depends on the past history of the dynamics. The dynamics can be made deterministic by setting $u = 0$. This includes the attractor dynamics popular in computational neuroscience [23, 3, 40].

## 4  Understanding the Learned Short-Run MCMC

### 4.1  Exponential Family and Moment Matching Estimator

An early version of EBM is the FRAME (Filters, Random field, And Maximum Entropy) model [55, 49, 54], which is an exponential family model, where the features are the responses from a bank of filters. The deep FRAME model [35] replaces the linear filters by the pre-trained ConvNet filters. This amounts to only learning the top layer weight parameters of the ConvNet. Specifically, $f_\theta(x) = \langle \theta, h(x) \rangle$, where $h(x)$ are the top-layer filter responses of a pre-trained ConvNet, and $\theta$ consists of the top-layer weight parameters. For such an $f_\theta(x)$, $\frac{\partial}{\partial \theta} f_\theta(x) = h(x)$. Then, the maximum likelihood estimator of $p_\theta$ is actually a moment matching estimator, i.e., $\mathbb{E}_{p_{\hat{\theta}_{\mathrm{MLE}}}}[h(x)] = \mathbb{E}_{p_{\mathrm{data}}}[h(x)]$. If we use the short-run MCMC learning algorithm, it will converge (assume convergence is attainable) to a moment matching estimator, i.e., $\mathbb{E}_{q_{\hat{\theta}_{\mathrm{MME}}}}[h(x)] = \mathbb{E}_{p_{\mathrm{data}}}[h(x)]$. Thus, the learned model $q_{\hat{\theta}_{\mathrm{MME}}}(x)$ is a valid estimator in that it matches to the data distribution in terms of sufficient statistics defined by the EBM.

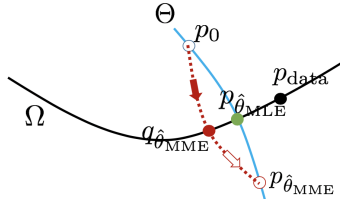

Figure 5: The blue curve illustrates the model distributions corresponding to different values of parameter $\theta$. The black curve illustrates all the distributions that match $p_{\mathrm{data}}$ (black dot) in terms of $\mathbb{E}[h(x)]$. The MLE $p_{\hat{\theta}_{\mathrm{MLE}}}$ (green dot) is the intersection between $\Theta$ (blue curve) and $\Omega$ (black curve). The MCMC (red dotted line) starts from $p_0$ (hollow blue dot) and runs toward $p_{\hat{\theta}_{\mathrm{MME}}}$ (hollow red dot), but the MCMC stops after $K$-step, reaching $q_{\hat{\theta}_{\mathrm{MME}}}$ (red dot), which is the learned short-run MCMC.

Consider two families of distributions: $\Omega = \{p : \mathbb{E}_p[h(x)] = \mathbb{E}_{p_{\mathrm{data}}}[h(x)]\}$, and $\Theta = \{p_\theta(x) = \exp(\langle \theta, h(x) \rangle)/Z(\theta), \forall \theta\}$. They are illustrated by two curves in Figure 5. $\Omega$ contains all the distributions that match the data distribution in terms of $\mathbb{E}[h(x)]$. Both $p_{\hat{\theta}_{\mathrm{MLE}}}$ and $q_{\hat{\theta}_{\mathrm{MME}}}$ belong to $\Omega$, and of course $p_{\mathrm{data}}$ also belongs to $\Omega$. $\Theta$ contains all the EBMs with different values of the parameter $\theta$. The uniform distribution $p_0$ corresponds to $\theta = 0$, thus $p_0$ belongs to $\Theta$.

The EBM under $\hat{\theta}_{\text{MME}}$, i.e., $p_{\hat{\theta}_{\text{MME}}}$ does not belong to $\Omega$, and it may be quite far from $p_{\hat{\theta}_{\text{MLE}}}$. In general, $\mathbb{E}_{p_{\hat{\theta}_{\text{MME}}}}[h(x)] \neq \mathbb{E}_{p_{\text{data}}}[h(x)]$, that is, the corresponding EBM does not match the data distribution as far as $h(x)$ is concerned. It can be much further from the uniform $p_0$ than $p_{\hat{\theta}_{\text{MLE}}}$ is from $p_0$, and thus $p_{\hat{\theta}_{\text{MME}}}$ may have a much lower entropy than $p_{\hat{\theta}_{\text{MLE}}}$.

Figure 5 illustrates the above idea. The red dotted line illustrates MCMC. Starting from $p_0$, $K$-step MCMC leads to $q_{\hat{\theta}_{\text{MME}}}(x)$. If we continue to run MCMC for infinite steps, we will get to $p_{\hat{\theta}_{\text{MME}}}$. Thus the role of $p_{\theta_{\text{MME}}}$ is to serve as an unreachable target to guide the $K$-step MCMC which stops at the mid-way $q_{\hat{\theta}_{\text{MME}}}$. One can say that the short-run MCMC is a wrong sampler of a wrong model, but it itself is a valid model because it belongs to $\Omega$.

The MLE $p_{\hat{\theta}_{\text{MLE}}}$ is the projection of $p_{\text{data}}$ onto $\Theta$. Thus it belongs to $\Theta$. It also belongs to $\Omega$ as can be seen from the maximum likelihood estimating equation. Thus it is the intersection of $\Omega$ and $\Theta$. Among all the distributions in $\Omega$, $p_{\hat{\theta}_{\text{MLE}}}$ is the closest to $p_0$. Thus it has the maximum entropy among all the distributions in $\Omega$.

The above duality between maximum likelihood and maximum entropy follows from the following fact. Let $\hat{p} \in \Theta \cap \Omega$ be the intersection between $\Theta$ and $\Omega$. $\Omega$ and $\Theta$ are orthogonal in terms of the Kullback-Leibler divergence. For any $p_\theta \in \Theta$ and for any $p \in \Omega$, we have the Pythagorean property [39]: $\text{KL}(p|p_\theta) = \text{KL}(p|\hat{p}) + \text{KL}(\hat{p}|p_\theta)$. See Appendix 7.1 for a proof. Thus (1) $\text{KL}(p_{\text{data}}|p_\theta) \geq \text{KL}(p_{\text{data}}|\hat{p})$, i.e., $\hat{p}$ is MLE within $\Theta$. (2) $\text{KL}(p|p_0) \geq \text{KL}(\hat{p}|p_0)$, i.e., $\hat{p}$ has maximum entropy within $\Omega$.

We can understand the learned $q_{\hat{\theta}_{\text{MME}}}$ from two Pythagorean results.

(1) Pythagorean for the right triangle formed by $q_0$, $q_{\hat{\theta}_{\text{MME}}}$, and $p_{\hat{\theta}_{\text{MLE}}}$,

$$\text{KL}(q_{\hat{\theta}_{\text{MME}}}|p_{\hat{\theta}_{\text{MLE}}}) = \text{KL}(q_{\hat{\theta}_{\text{MME}}}|p_0) - \text{KL}(p_{\hat{\theta}_{\text{MLE}}}|p_0) = H(p_{\hat{\theta}_{\text{MLE}}}) - H(q_{\hat{\theta}_{\text{MME}}}), \qquad (8)$$

where $H(p) = -\mathbb{E}_p[\log p(x)]$ is the entropy of $p$. See Appendix 7.1. Thus we want the entropy of $q_{\hat{\theta}_{\text{MME}}}$ to be high in order for it to be a good approximation to $p_{\hat{\theta}_{\text{MLE}}}$. Thus for small $K$, it is important to let $p_0$ be the uniform distribution, which has the maximum entropy.

(2) Pythagorean for the right triangle formed by $p_{\hat{\theta}_{\text{MME}}}$, $q_{\hat{\theta}_{\text{MME}}}$, and $p_{\hat{\theta}_{\text{MLE}}}$,

$$\text{KL}(q_{\hat{\theta}_{\text{MME}}}|p_{\hat{\theta}_{\text{MME}}}) = \text{KL}(q_{\hat{\theta}_{\text{MME}}}|p_{\hat{\theta}_{\text{MLE}}}) + \text{KL}(p_{\hat{\theta}_{\text{MLE}}}|p_{\hat{\theta}_{\text{MME}}}). \qquad (9)$$

For fixed $\theta$, as $K$ increases, $\text{KL}(q_\theta|p_\theta)$ decreases monotonically [7]. The smaller $\text{KL}(q_{\hat{\theta}_{\text{MME}}}|p_{\hat{\theta}_{\text{MME}}})$ is, the smaller $\text{KL}(q_{\hat{\theta}_{\text{MME}}}|p_{\hat{\theta}_{\text{MLE}}})$ and $\text{KL}(p_{\hat{\theta}_{\text{MLE}}}|p_{\hat{\theta}_{\text{MME}}})$ are. Thus, it is desirable to use large $K$ as long as we can afford the computational cost, to make both $q_{\hat{\theta}_{\text{MME}}}$ and $p_{\hat{\theta}_{\text{MME}}}$ close to $p_{\hat{\theta}_{\text{MLE}}}$.

## 4.2 General ConvNet-EBM and Generalized Moment Matching Estimator

For a general ConvNet $f_\theta(x)$, the learning algorithm based on short-run MCMC solves the following estimating equation: $\mathbb{E}_{q_\theta}\left[\frac{\partial}{\partial \theta} f_\theta(x)\right] = \mathbb{E}_{p_{\text{data}}}\left[\frac{\partial}{\partial \theta} f_\theta(x)\right]$, whose solution is $\hat{\theta}_{\text{MME}}$, which can be considered a generalized moment matching estimator that in general solves the following estimating equation: $\mathbb{E}_{q_\theta}[h(x, \theta)] = \mathbb{E}_{p_{\text{data}}}[h(x, \theta)]$, where we generalize $h(x)$ in the original moment matching estimator to $h(x, \theta)$ that involves both $x$ and $\theta$. For our learning algorithm, $h(x, \theta) = \frac{\partial}{\partial \theta} f_\theta(x)$. That is, the learned $q_{\hat{\theta}_{\text{MME}}}$ is still a valid estimator in the sense of matching to the data distribution. The above estimating equation can be solved by Robbins-Monro's stochastic approximation [42], as long as we can generate independent fair samples from $q_\theta$.

In classical statistics, we often assume that the model is correct, i.e., $p_{\text{data}}$ corresponds to a $q_{\theta_{\text{true}}}$ for some true value $\theta_{\text{true}}$. In that case, the generalized moment matching estimator $\hat{\theta}_{\text{MME}}$ follows an asymptotic normal distribution centered at the true value $\theta_{\text{true}}$. The variance of $\hat{\theta}_{\text{MME}}$ depends on the choice of $h(x, \theta)$. The variance is minimized by the choice $h(x, \theta) = \frac{\partial}{\partial \theta} \log q_\theta(x)$, which corresponds to the maximum likelihood estimate of $q_\theta$, and which leads to the Cramer-Rao lower bound and Fisher information. See Appendix 7.2 for a brief explanation.

$\frac{\partial}{\partial \theta} \log p_\theta(x) = \frac{\partial}{\partial \theta} f_\theta(x) - \frac{\partial}{\partial \theta} \log Z(\theta)$ is not equal to $\frac{\partial}{\partial \theta} \log q_\theta(x)$. Thus the learning algorithm will not give us the maximum likelihood estimate of $q_\theta$. However, the validity of the learned $q_\theta$ does

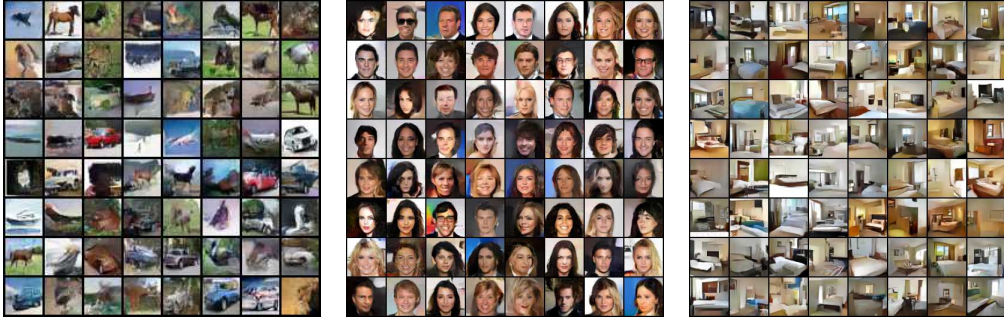

Figure 6: Generated samples for $K = 100$ MCMC steps. From left to right: (1) CIFAR-10 ($32 \times 32$), (2) CelebA ($64 \times 64$), (3) LSUN Bedroom ($64 \times 64$).

| Model | CIFAR-10 | CelebA | LSUN Bedroom |
|---|---|---|---|
| | IS | FID | FID |
| VAE [28] | 4.28 | 79.09 | 183.18 |
| DCGAN [41] | 6.16 | 32.71 | 54.17 |
| Ours | **6.21** | **23.02** | **44.16** |

(a) IS and FID scores for generated examples.

| Model | CIFAR-10 | CelebA | LSUN Bedroom |
|---|---|---|---|
| | MSE | MSE | MSE |
| VAE [28] | 0.0421 | 0.0341 | 0.0440 |
| DCGAN [41] | 0.0407 | 0.0359 | 0.0592 |
| Ours | **0.0387** | **0.0271** | **0.0272** |

(b) Reconstruction error (MSE per pixel).

Table 1: Quality of synthesis and reconstruction for CIFAR-10 ($32 \times 32$), CelebA ($64 \times 64$), and LSUN Bedroom ($64 \times 64$). The number of features $n_f$ is 128, 64, and 64, respectively, and $K = 100$.

not require $h(x, \theta)$ to be $\frac{\partial}{\partial \theta} \log q_\theta(x)$. In practice, one can never assume that the model is true. As a result, the optimality of the maximum likelihood may not hold, and there is no compelling reason that we must use MLE.

The relationship between $p_{\text{data}}, q_{\hat{\theta}_{\text{MME}}}, p_{\hat{\theta}_{\text{MME}}}$, and $p_{\hat{\theta}_{\text{MLE}}}$ may still be illustrated by Figure 5, although we need to modify the definition of $\Omega$.

## 5 Experimental Results

In this section, we will demonstrate (1) realistic synthesis, (2) smooth interpolation, (3) faithful reconstruction of observed examples, and, (4) the influence of hyperparameters. $K$ denotes the number of MCMC steps in equation (4). $n_f$ denotes the number of output features maps in the first layer of $f_\theta$. See Appendix for additional results.

We emphasize the simplicity of the algorithm and models, see Appendix 7.3 and 7.4, respectively.

### 5.1 Fidelity

We evaluate the fidelity of generated examples on various datasets, each reduced to $40,000$ observed examples. Figure 6 depicts generated samples for various datasets with $K = 100$ Langevin steps for both training and evaluation. For CIFAR-10 we set the number of features $n_f = 128$, whereas for CelebA and LSUN we use $n_f = 64$. We use $200,000$ iterations of model updates, then gradually decrease the learning rate $\eta$ and injected noise $\varepsilon_i \sim \text{N}(0, \sigma^2 I)$ for observed examples. Table 1 (a) compares the Inception Score (IS) [45, 4] and Fréchet Inception Distance (FID) [20] with Inception v3 classifier [47] on $40,000$ generated examples. Despite its simplicity, short-run MCMC is competitive.

### 5.2 Interpolation

We demonstrate interpolation between generated examples. We follow the procedure outlined in Section 3.4. Let $x_\rho = M_\theta(z_\rho)$ where $M_\theta$ to denotes the $K$-step gradient descent with $K = 100$. Figure 3 illustrates $x_\rho$ for a sequence of $\rho \in [0, 1]$ on CelebA. The interpolation appears smooth and the intermediate samples resemble realistic faces. The interpolation experiment highlights that the short-run MCMC does not mix, which is in fact an advantage instead of a disadvantage. The interpolation ability goes far beyond the capacity of EBM and convergent MCMC.

|  | $K$ | | | | | |
|---|---|---|---|---|---|---|
|  | 5 | 10 | 25 | 50 | 75 | 100 |
| $\sigma$ | 0.15 | 0.1 | 0.05 | 0.04 | 0.03 | **0.03** |
| FID | 213.08 | 182.5 | 92.13 | 68.28 | 65.37 | **63.81** |
| IS | 2.06 | 2.27 | 4.06 | 4.82 | 4.88 | **4.92** |
| $\|\frac{\partial}{\partial x}f_\theta(x)\|_2$ | 7.78 | 3.85 | 1.76 | 0.97 | 0.65 | **0.49** |

Table 2: Influence of number of MCMC steps $K$ on models with $n_f = 32$ for CIFAR-10 ($32 \times 32$).

|  | $\sigma$ | | | | | |
|---|---|---|---|---|---|---|
|  | 0.10 | 0.08 | 0.06 | 0.05 | 0.04 | 0.03 |
| FID | 132.51 | 117.36 | 94.72 | 83.15 | 65.71 | **63.81** |
| IS | 4.05 | 4.20 | 4.63 | 4.78 | 4.83 | **4.92** |

|  | $n_f$ | | |
|---|---|---|---|
|  | 32 | 64 | 128 |
| FID | 63.81 | 46.61 | **44.50** |
| IS | 4.92 | 5.49 | **6.21** |

(a) Influence of additive noise $\varepsilon_i \sim \mathrm{N}(0, \sigma^2 I)$.  (b) Influence of model complexity $n_f$.

Table 3: Influence of noise and model complexity with $K = 100$ for CIFAR-10 ($32 \times 32$).

## 5.3 Reconstruction

We demonstrate reconstruction of observed examples. For short-run MCMC, we follow the procedure outlined in Section 3.4. For an observed image $x$, we reconstruct $x$ by running gradient descent on the least squares loss function $L(z) = \|x - M_\theta(z)\|^2$, initializing from $z_0 \sim p_0(z)$, and iterates $z_{t+1} = z_t - \eta_t L'(z_t)$. For VAE, reconstruction is readily available. For GAN, we perform Langevin inference of latent variables [19, 50]. Figure 4 depicts faithful reconstruction. Table 1 (b) illustrates competitive reconstructions in terms of MSE (per pixel) for $1,000$ observed leave-out examples. Again, the reconstruction ability of the short-run MCMC is due to the fact that it is not mixing.

## 5.4 Influence of Hyperparameters

**MCMC Steps.** Table 2 depicts the influence of varying the number of MCMC steps $K$ while training on synthesis and average magnitude $\|\frac{\partial}{\partial x}f_\theta(x)\|_2$ over $K$-step Langevin (4). We observe: (1) the quality of synthesis decreases with decreasing $K$, and, (2) the shorter the MCMC, the colder the learned EBM, and the more dominant the gradient descent part of the Langevin. With small $K$, short-run MCMC fails "gracefully" in terms of synthesis. A choice of $K = 100$ appears reasonable.

**Injected Noise.** To stabilize training, we smooth $p_{\text{data}}$ by injecting additive noises $\varepsilon_i \sim \mathrm{N}(0, \sigma^2 I)$ to observed examples $x_i \leftarrow x_i + \varepsilon_i$. Table 3 (a) depicts the influence of $\sigma^2$ on the fidelity of negative examples in terms of IS and FID. That is, when lowering $\sigma^2$, the fidelity of the examples improves. Hence, it is desirable to pick smallest $\sigma^2$ while maintaining the stability of training. Further, to improve synthesis, we may gradually decrease the learning rate $\eta$ and anneal $\sigma^2$ while training.

**Model Complexity.** We investigate the influence of the number of output features maps $n_f$ on generated samples with $K = 100$. Table 3 (b) summarizes the quality of synthesis in terms of IS and FID. As the number of features $n_f$ increases, so does the quality of the synthesis. Hence, the quality of synthesis may scale with $n_f$ until the computational means are exhausted.

## 6 Conclusion

Despite our focus on short-run MCMC, we do not advocate abandoning EBM all together. On the contrary, we ultimately aim to learn valid EBM [38]. Hopefully, the non-convergent short-run MCMC studied in this paper may be useful in this endeavor. It is also our hope that our work may help to understand the learning of attractor dynamics popular in neuroscience.

**Acknowledgments**

The work is supported by DARPA XAI project N66001-17-2-4029; ARO project W911NF1810296; and ONR MURI project N00014-16-1-2007; and XSEDE grant ASC170063. We thank Prof. Stu Geman, Prof. Xianfeng (David) Gu, Diederik P. Kingma, Guodong Zhang, and Will Grathwohl for helpful discussions.

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
