[Supplementary Material]

# On Learning Non-Convergent Non-Persistent Short-Run MCMC Toward Energy-Based Model

## 1 Supplementary Experimental Results

Figure 1: **Synthesis by short-run MCMC**: Generating synthesized examples by running 100 steps of Langevin dynamics initialized from uniform noise for CelebA ($128 \times 128$).

### 1.1 Super-Resolution

We demonstrate informative initialization by means of super-resolution synthesis. Consider the problem of super-resolution in which we aim to generate a high-resolution image given a low-resolution image. In training, we run a small number of Langevin steps from the low-resolution image. That is, the model is still $M_\theta p_0$, except $p_0$ is lower resolution, not uniform. Figure 2 demonstrates up-sampling from $4 \times 4$, $8 \times 8$, and $16 \times 16$ to $64 \times 64$ pixels. While the choice of an informative initialization $p_0$ is less desirable, the super-resolution experiment demonstrates that short-run MCMC can be generalized to various tasks.

Figure 2: Super-resolution using Langevin with $K = 100$ and $n_f = 32$ on CelebA ($64 \times 64$). The random seed is varied to generate four sets of samples for each initialization. Top row: $4 \times 4$ to $64 \times 64$. Middle row: $8 \times 8$ to $64 \times 64$. Bottom row: $16 \times 16$ to $64 \times 64$.

Figure 3: Interpolation between generated examples with $K = 100$ and $n_f = 32$ on CelebA ($64 \times 64$). The transition depicts the sequence $M_\theta(z_\rho)$ with interpolated noise $z_\rho = \rho z_1 + \sqrt{1 - \rho^2} z_2$ where $\rho \in [0, 1]$. Left: $z_1 \sim p_0$. Right: $z_2 \sim p_0$.

Figure 4: Reconstruction of observed examples with $K = 100$ and $n_f = 32$ on CelebA ($64 \times 64$). The transition depicts $M_\theta(z_t)$ over time $t$ from random initialization $t = 0$ to reconstruction $t = 200$. Left: Random initialization. Right: Observed examples.

 ## 2  Appendix

 ### 2.1  Proof of Pythagorean Identity

12  For $p \in \Omega$, let $\mathbb{E}_p[h(x)] = \mathbb{E}_{p_{\text{data}}}[h(x)] = \hat{h}$.

$$\begin{aligned} \text{KL}(p|p_\theta) &= \mathbb{E}_p[\log p(x) - \langle \theta, h(x) \rangle + \log Z(\theta)] \quad (1) \\ &= -H(p) - \langle \theta, \hat{h} \rangle + \log Z(\theta), \quad (2) \end{aligned}$$

13  where $H(p) = -\mathbb{E}_p[\log p(x)]$ is the entropy of $p$.

14  For $\hat{p} \in \Omega \cap \Theta$,

$$\text{KL}(\hat{p}|p_\theta) = -H(\hat{p}) - \langle \theta, \hat{h} \rangle + \log Z(\theta). \quad (3)$$

$$\begin{aligned} \text{KL}(p|\hat{p}) &= \mathbb{E}_p[\log p(x)] - \mathbb{E}_p[\log \hat{p}(x)] \quad (4) \\ &= \mathbb{E}_p[\log p(x)] - \mathbb{E}_{\hat{p}}[\log \hat{p}(x)] \quad (5) \\ &= -H(p) + H(\hat{p}). \quad (6) \end{aligned}$$

16  Thus $\text{KL}(p|p_\theta) = \text{KL}(p|\hat{p}) + \text{KL}(\hat{p}|p_\theta)$.

### 2.2  Estimating Equation and Cramer-Rao Theory

18  For a model $q_\theta$, we can estimate $\theta$ by solving the estimating equation $\mathbb{E}_{q_\theta}[h(x, \theta)] = \frac{1}{n} \sum_{i=1}^n h(x_i, \theta)$.
19  Assume the solution exists and let it be $\hat{\theta}$. Assume there exists $\theta_{\text{true}}$ so that $p_{\text{data}} = q_{\theta_{\text{true}}}$. Let $c(\theta) =$
20  $\mathbb{E}_{q_\theta}[h(x, \theta)]$. We can change $h(x, \theta) \leftarrow h(x, \theta) - c(\theta)$. Then $\mathbb{E}_{q_\theta}[h(x, \theta)] = 0 \, \forall \theta$, and the estimating
21  equation becomes $\frac{1}{n} \sum_{i=1}^n h(x_i, \theta) = 0$. A Taylor expansion around $\theta_{\text{true}}$ gives us the asymptotic
22  linear equation $\frac{1}{n} \sum_{i=1}^n [h(x_i, \theta_{\text{true}}) + h'(x_i, \theta_{\text{true}})(\theta - \theta_{\text{true}})] = 0$, where $h'(x, \theta) = \frac{\partial}{\partial \theta} h(x, \theta)$. Thus
23  the estimate $\hat{\theta} = \theta_{\text{true}} - \left[ \frac{1}{n} \sum_{i=1}^n h'(x_i, \theta_{\text{true}}) \right]^{-1} \left[ \frac{1}{n} \sum_{i=1}^n h(x_i, \theta_{\text{true}}) \right]$, i.e., one-step Newton-Raphson
24  update from $\theta_{\text{true}}$. Since $\mathbb{E}_{q_\theta}[h(x, \theta)] = 0$ for any $\theta$, including $\theta_{\text{true}}$, the estimator $\hat{\theta}$ is asymptotically
25  unbiased. The Cramer-Rao theory establishes that $\hat{\theta}$ has an asymptotic normal distribution, $\sqrt{n}(\hat{\theta} -$
26  $\theta_{\text{true}}) \sim N(0, V)$, where $V = \mathbb{E}_{\theta_{\text{true}}}[h'(x, \theta_{\text{true}}]^{-1} \text{Var}_{\theta_{\text{true}}}[h(x; \theta_{\text{true}})] \mathbb{E}_{\theta_{\text{true}}}[h'(x, \theta_{\text{true}}]^{-1}$. $V$ is minimized
27  if we take $h(x, \theta) = \frac{\partial}{\partial \theta} \log q_\theta(x)$, which leads to the maximum likelihood estimating equation, and
28  the corresponding $V = I(\theta_{\text{true}})^{-1}$, where $I$ is the Fisher information.

 ## 2.3 Code

```python
import torch as t, torch.nn as nn
import torchvision as tv, torchvision.transforms as tr

seed = 1
im_sz = 32
sigma = 3e-2 # decrease until training is unstable
n_ch = 3
m = 8**2
K = 100
n_f = 64 # increase until compute is exhausted
n_i = 10**5

t.manual_seed(seed)
if t.cuda.is_available():
    t.cuda.manual_seed_all(seed)

device = t.device('cuda' if t.cuda.is_available() else 'cpu')

class F(nn.Module):
    def __init__(self, n_c=n_ch, n_f=n_f, l=0.2):
        super(F, self).__init__()
        self.f = nn.Sequential(
            nn.Conv2d(n_c, n_f, 3, 1, 1),
            nn.LeakyReLU(l),
            nn.Conv2d(n_f, n_f * 2, 4, 2, 1),
            nn.LeakyReLU(l),
            nn.Conv2d(n_f * 2, n_f * 4, 4, 2, 1),
            nn.LeakyReLU(l),
            nn.Conv2d(n_f * 4, n_f * 8, 4, 2, 1),
            nn.LeakyReLU(l),
            nn.Conv2d(n_f * 8, 1, 4, 1, 0))

    def forward(self, x):
        return self.f(x).squeeze()

f = F().to(device)

transform = tr.Compose([tr.Resize(im_sz), tr.ToTensor(), tr.Normalize((.5, .5, .5), (.5, .5, .5))])
p_d = t.stack([x[0] for x in tv.datasets.CIFAR10(root='data/cifar10', transform=transform)]).to(device)
noise = lambda x: x + sigma * t.randn_like(x)
def sample_p_d():
    p_d_i = t.LongTensor(m).random_(0, p_d.shape[0])
    return noise(p_d[p_d_i]).detach()

sample_p_0 = lambda: t.FloatTensor(m, n_ch, im_sz, im_sz).uniform_(-1, 1).to(device)
def sample_q(K=K):
    x_k = t.autograd.Variable(sample_p_0(), requires_grad=True)
    for k in range(K):
        f_prime = t.autograd.grad(f(x_k).sum(), [x_k], retain_graph=True)[0]
        x_k.data += f_prime + 1e-2 * t.randn_like(x_k)
    return x_k.detach()

sqrt = lambda x: int(t.sqrt(t.Tensor([x])))
plot = lambda p, x: tv.utils.save_image(t.clamp(x, -1., 1.), p, normalize=True, nrow=sqrt(m))

optim = t.optim.Adam(f.parameters(), lr=1e-4, betas=[.9, .999])

for i in range(n_i):
    x_p_d, x_q = sample_p_d(), sample_q()
    L = f(x_p_d).mean() - f(x_q).mean()
    optim.zero_grad()
    (-L).backward()
    optim.step()

    if i % 100 == 0:
        print('{:>6d} f(x_p_d)={:>14.9f} f(x_q)={:>14.9f}'.format(i, f(x_p_d).mean(), f(x_q).mean()))
        plot('x_q_{:>06d}.png'.format(i), x_q)
```

## 2.4 Model Architecture

We use the following notation. Convolutional operation $conv(n)$ with $n$ output feature maps and bias term. Leaky-ReLU nonlinearity *LReLU* with default leaky factor 0.2. We set $n_f \in \{32, 64, 128\}$.

| Energy-based Model ($32 \times 32 \times 3$) | | |
|---|---|---|
| Layers | In-Out Size | Stride |
| Input | $32 \times 32 \times 3$ | |
| $3 \times 3$ conv($n_f$), LReLU | $32 \times 32 \times n_f$ | 1 |
| $4 \times 4$ conv($2*n_f$), LReLU | $16 \times 16 \times (2*n_f)$ | 2 |
| $4 \times 4$ conv($4*n_f$), LReLU | $8 \times 8 \times (4*n_f)$ | 2 |
| $4 \times 4$ conv($8*n_f$), LReLU | $4 \times 4 \times (8*n_f)$ | 2 |
| $4 \times 4$ conv(1) | $1 \times 1 \times 1$ | 1 |

Table 1: Network structures ($32 \times 32 \times 3$).

| Energy-based Model ($64 \times 64 \times 3$) | | |
|---|---|---|
| Layers | In-Out Size | Stride |
| Input | $64 \times 64 \times 3$ | |
| $3 \times 3$ conv($n_f$), LReLU | $64 \times 64 \times n_f$ | 1 |
| $4 \times 4$ conv($2*n_f$), LReLU | $32 \times 32 \times (2*n_f)$ | 2 |
| $4 \times 4$ conv($4*n_f$), LReLU | $16 \times 16 \times (4*n_f)$ | 2 |
| $4 \times 4$ conv($8*n_f$), LReLU | $8 \times 8 \times (8*n_f)$ | 2 |
| $4 \times 4$ conv($8*n_f$), LReLU | $4 \times 4 \times (8*n_f)$ | 2 |
| $4 \times 4$ conv(1) | $1 \times 1 \times 1$ | 1 |

Table 2: Network structures ($64 \times 64 \times 3$).

| Energy-based Model ($128 \times 128 \times 3$) | | |
|---|---|---|
| Layers | In-Out Size | Stride |
| Input | $128 \times 128 \times 3$ | |
| $3 \times 3$ conv($n_f$), LReLU | $128 \times 128 \times n_f$ | 1 |
| $4 \times 4$ conv($2*n_f$), LReLU | $64 \times 64 \times (2*n_f)$ | 2 |
| $4 \times 4$ conv($4*n_f$), LReLU | $32 \times 32 \times (4*n_f)$ | 2 |
| $4 \times 4$ conv($8*n_f$), LReLU | $16 \times 16 \times (8*n_f)$ | 2 |
| $4 \times 4$ conv($8*n_f$), LReLU | $8 \times 8 \times (8*n_f)$ | 2 |
| $4 \times 4$ conv($8*n_f$), LReLU | $4 \times 4 \times (8*n_f)$ | 2 |
| $4 \times 4$ conv(1) | $1 \times 1 \times 1$ | 1 |

Table 3: Network structures ($128 \times 128 \times 3$).