[Reviews · NeurIPS 2019]

Reviewer 1



Originality: The main and novel contribution of this paper is showing how to exploit/embrace the non-convergence of MCMC to learn interesting models. Related work is well cited, and it is clear how it differs from previous approach like contrastive divergence and moment matching GANs. There are also interesting connections to energy based dynamics to models used in neuroscience. Quality: The authors have come up with an interesting novel strategy of embracing what was seen as undesired property (non-convergent etc MCMC), and built a complete and interesting work exploring the results of said strategy . The paper appears technically sound and it should be possible to recreate their model and results from the descriptions in the paper (ignoring the fact that the code is already provided). The paper also does a good job of exploring the connection to the related work. Minor correction: Line 136 - "a deterministic" -> "made deterministic". Clarity: The paper was clearly written and understandable.. A minor improvement for Section 4 would be to start by clearly stating that the FRAME model would will be examined first as stronger statements can be made in the restricted regime. Significance: The significance of the paper is potentially quite high as it may unlock deeper connections between EBM, MCMC and more general CNN approaches. It is also a quite simple algorithm, but still exhibits competitive results. It might also yield insights into why existing methods like CD work despite not having convergence in their MCMC chains. See also the contributions section.

Reviewer 2



The highlighted phenomenon (the convergence of a short-run MCMC while training EBMs) seems to be novel and very interesting. The conventional wisdom is that a simple MCMC algorithm like Langevin dynamics would take a long time to converge close to the stationary distribution of the EBM when initialized far from it. The paper argues that in fact if the EBM is trained by generating negative samples from a short-run MCMC, then the short-run MCMC chain would in fact converge close to the data distribution (the authors argue that the "closeness" is related to moment matching). The theoretical argument for explaining this phenomenon seems suggestive, but ultimately didn't convince the reviewer (even convergence of the algorithm seems to be not explained, and section 4.2 seems particularly weak - it's not clear what the "generalized moment matching objective" is trying to achieve). However the empirical evidence for the convergence of short-run MCMC in EBMs seems very compelling - the training procedure for the model is significantly simpler than other procedures used to train EBMs, yet produces highly competitive results on several image datasets. There is some evidence for coverage of the distribution, which is a concern for models not trained with the MLE objective (the evidence is the reconstructions of held-out data points). It would be great to see actual log-likelihood estimates under the short-run MCMC model (not the EBM, the directed model that is K steps of Langevin dynamics) by training a recognition network and computing a lower bound on the likelihood similar to "Accurate and Conservative Estimates of MRF Log-likelihood using Reverse Annealing, Burda et al, 2014". The submission is written very clearly, and the code is provided and easy to read. The reviewer believes the work is significant, as it highlights a new phenomenon in training energy-based models not yet explored in the relevant literature. While the theoretical explanation didn't convince the reviewer, I believe that future work will attempt to explain the phenomenon of short-run MCMC convergence more rigorously.

Reviewer 3



the paper provides an interesting statistical justification of the short-run MCMC. it views the short-run MCMC as a generative model, with initial image as the latent variable, uniform noise as prior, and the Langevin dynamics as network. this is an interesting formulation. I have three major comments. one is on why short-run MCMC can reconstruct the observed images and interpolating different images. is it due to the "short-run" property that allows short-run MCMC for good reconstruction? it is due to the choice of a fixed K? how does the theoretical argument in the paper help justify this? while the theoretical argument says that the short-run MCMC is preferrable because the short-run MCMC does not need to have the EBM as the stationary distribution. however the empirical results in tables 2 and 3 still favor large K. does it mean short-run MCMC still favors stationary distribution? the short-run MCMC also appears closely related to Monte Carlo EM, where it is common to run MCMC for a fixed number of steps in the E-step without reaching a stationary distribution. does the theoretical results of Monte Carlo EM directly apply to short-run MCMC here?

[Author Response · NeurIPS 2019]

**On Learning Non-Convergent Non-Persistent Short-Run MCMC Toward Energy-Based Model.**

**Reply to Reviewer 2:** Thank you for the insightful and comprehensive summary of our work.

**Q1: About training time. A1:** As you have pointed out, each iteration requires computing $K$ derivatives of the CNN.
As an example, $100,000$ model parameter updates for $64 \times 64$ CelebA with $K = 100$ and $n_f = 64$ on 4 Titan Xp
GPUs take 16 hours. We will add such information in revision.
**Q2: Dynamic $K$. A2:** Following your advice, we conducted experiments with random $K \in [100, 120]$ for training.
We can still learn short-run MCMC successfully. This corresponds to residual network with random number of layers.
**Q3: Using different $K$ for training and sampling. A3:** Following your advice, we conducted an experiment on
CelebA, where we train the model with $K_1$ and test the trained model by running MCMC with $K_2$ steps. The Figures
below depict training with $K_1 \in \{100, 50\}$ and varied $K_2$ for sampling. Note, over-saturation occurs for $K_2 > K_1$.

Transition with $K_1 = 100$ for training and varying $K_2$ for sampling.     Transition with $K_1 = 50$ for training.

$K_2 = 0$   20   40   60   80   100   120   140   160   180   200        $K_2 = 25$   50   100

**Reply to Reviewer 3:** Thank you for the insightful comments.

**Q1: About theoretical justification. A1:** Your comment is well taken, and we shall continue to try our best on
theoretical understanding. About the generalized moment matching estimator, classical statistical theory shows that the
estimator is asymptotically unbiased, and the variance of the estimator can also be derived (see Supplementary 2.2).
The convergence of the learning algorithm follows Robbins-Monro stochastic approximation because $q_\theta(x)$ can be
sampled exactly.
**Q2: About computing log-likelihood. A2:** Following your advice, we computed log-likelihood. The prior model is
$z \sim p_0(z)$ which is the uniform distribution. After training, we learn the dynamics $x = M_\theta(z)$, where $M_\theta$ consists
of $K$-steps of gradient descent dynamics (the noise term of the Langevin dynamics is negligible compared to the
gradient term after learning). Under this flow dynamics $p(x) = p_0(z)/\det(\partial M_\theta(z)/\partial z))$ (where we used GELU, a
differentiable version of ReLU). For each observed $x$, we can obtain $z$ as described in Section 3.4. Then we compute
the Jacobian and its determinant as the product of the eigenvalues of the Jacobian. In our preliminary results, the
log-likelihood computation is feasible for images of size $32 \times 32 \times 3$. For a small batch of 16 images, we have obtained
a rough, preliminary estimate average $4.12$ number of bits per data dimension. We will include log-likelihood results in
revision. We shall also try to implement Burda et al. (thanks for the reference).
**Q3: 1D and 2D toy examples. A3:** Following your advice, we did 1D and 2D experiments. We plot the density and
log-density of the true model, the learned EBM, and the kernel density estimate (KDE, like histogram) of the MCMC
samples. The density of the MCMC samples matches the true density closely. The learned energy captures the modes of
the true density, but is of a much bigger scale, so that the learned EBM density is of much lower entropy or temperature
(so that the density focuses on the global energy minimum). This is consistent with our theoretical understanding.

**Reply to Reviewer 4:** Thank you for the interesting questions.

**Q1: About ability to reconstruct. A1:** You are right. It is due to short-run non-mixing, i.e., different starting point
$z$ leads to different $x$ after $K$-step MCMC, and $K$ is fixed, so that a mapping is well-defined $z = M_\theta(x)$, where $M_\theta$
consists of $K$-steps of gradient descent dynamics (the noise term of the Langevin dynamics is negligible compared to
the gradient term after learning). $M_\theta$ can be considered a flow model.
**Q2: Large $K$ is favored. A2:** We believe larger $K$ achieves improved synthesis results because larger $K$ leads to a
better learned flow model $M_\theta(z)$ (as a residual network with more layers). Our experience suggests that increasing $K$
much further beyond 100 does not lead to much improved results.
**Q3: Monte Carlo EM. A3:** Our short-run MCMC always initializes from noise images sampled from the uniform
distribution. In Monte Carlo EM, one may initialize $K$-step MCMC that samples from the posterior distribution of the
latent variables from the currently generated samples, i.e., running the so-called persistent chains with warm start. Of
course, one may also initialize from the same noise distribution, i.e., cold start. In that case, the short-run MCMC may
be interpreted as a variational inference model. We have used this method to learn a latent EBM similar to Boltzmann
machine but with continuous latent variables, where we use short-run MCMC for both inference and synthesis.

[Meta-Review · NeurIPS 2019]

A new energy-based generative model for images is proposed. The paper suggests to run Langevin dynamics in the data domain to create artificial samples, and updating the model parameters based on these synthesized images in an 'analysis by synthesis' framework. The generative model allows for unconditional generation and interpolation. It is interesting that short-run MCMC can be used in this context despite not being converged. The effect of the hyperparameter K (number of MCMC steps) could have been more explored. The theoretical part has weaknesses and should be improved in the final version. Overall, this is an interesting piece of work.